# The Role of Vulture (Accipitriformes) Cutaneous Microbiota in Infectious Disease Protection

**DOI:** 10.3390/microorganisms13040898

**Published:** 2025-04-14

**Authors:** Miriam Lobello, Roberto Bava, Fabio Castagna, Francesca Daniela Sotgiu, Fiammetta Berlinguer, Bruno Tilocca

**Affiliations:** 1Department of Health Science, University of Catanzaro, 88100 Catanzaro, Italyroberto.bava@unicz.it (R.B.); fabiocastagna@unicz.it (F.C.); 2Department of Veterinary Medicine, University of Sassari, 07100 Sassari, Italy; fdsotgiu@uniss.it (F.D.S.); berling@uniss.it (F.B.)

**Keywords:** microbiome, skin microbial community, necrophagic animals, microbial competition

## Abstract

Vultures (Accipitriformes), as obligate scavengers, are regularly exposed to a diverse array of pathogens present in decomposing carcasses. Nevertheless, they exhibit a remarkable ability to resist infections, suggesting a crucial role of skin microbiota in host defense. The microbial communities residing on necrophagic birds’ skin create a protective barrier through competitive interactions, antimicrobial compound production, and immunity priming. Additionally, vultures contribute to ecosystem balance by reducing the spread of infectious agents. However, they may also serve as vectors for antimicrobial resistance (AMR) due to their exposure to contaminated food sources. Understanding the dynamics of their microbiota can provide valuable insights into host–microbe interactions, wildlife conservation, and public health. This review examines the composition and functional significance of vulture cutaneous microbiota. Specifically, it explores the role of necrophagic birds’ skin microbiota in pathogen exclusion, immune system modulation, and environmental adaptation, with the aim of suggesting further research routes, besides clarifying the ecological implications of such birds.

## 1. Introduction

Necrophagic animals are exposed to numerous pathogens, many of which are of human relevance such as *Salmonella enterica* subsp. *enterica serovar Enteritidis*, *Escherichia* spp. *i*, and *Campylobacter* spp. [1,2]. Particular attention is directed to such microbial specimens, acknowledging their broad host spectrum, zoonotic potential, and association with antibiotic resistance [2]. Also, by feeding on decomposing animals, necrophagous animals increase their risk of exposure to toxins and/or microbial catabolites resulting from the decomposition processes of the carcasses [3]. In addition, the dietary style of necrophagic birds leads to the contamination of various anatomical regions, especially the head and neck, facilitating the vehiculation of such microorganisms of One Health relevance [1,3]. On the other hand, concerns remain unsolved about how these animals cope with such biological threats. In recent years, attention has been focused on the importance of the microbiota, highlighting the role of the skin microbiota in determining necrophagous health status [4]. In the present review, we aim to provide the state of the art of the latest studies available in the literature, aiming to characterize the structural and functional aspects of cutaneous microbiota and their impact on infectious diseases in necrophagic animals. The vultures of the *Cathartidae* family are employed as reference models for necrophagic animals and a discussion on its role in pathogen dissemination is provided. Also, we review how cutaneous microbiota may represent a health risk for both human and animal populations, including its potential implications in the transmission of antibiotic-resistant traits.

## 2. Vultures

Vultures are among the largest scavenging animals known, in terms of size [5]. They are obligate scavenging raptors, traditionally considered symbols of death, purification, and rebirth, present in the culture of various civilizations for centuries and valued in daily life for their role as a “clean-up bird” [5,6,7]. Although all vulture species are obligatory scavengers, varying intensity in scavenging and the diverse preferences for the components of the animal carcasses favor the coexistence of three main scavenger groups, namely “gulpers”, “rippers”, and “scrappers” [1]. These are morphologically distinguished, particularly by their skull, among other features [1]. Gulpers feed on soft tissues, removing portions of tender meat by inserting their heads and necks into skin openings and swallowing pieces while lifting their heads [1]. Rippers feed on tougher superficial tissues like skin, muscles, and tendons by tearing pieces using their feet as anchors [1]. Scrappers exhibit a feeding behavior similar to that of chickens, pecking small pieces of meat located on the surface and around carcasses [1].

### 2.1. Classification of Vultures

Twenty-three species of vultures are distinguished worldwide, residing in various biomes from the Amazon rainforest and East African savannas to the Sahara Desert and the high Himalayas [5]. There are two main groups of vultures, namely (i) Old World vultures, including 16 species (griffons and vultures) belonging to the *Accipitridae* family, distributed across Europe, Africa, and Asia, and (ii) New World vultures, including seven species (condors) belonging to the *Cathartidae* family, located in North and South America. The former are more closely related to eagles and falcons, while the latter are more akin to storks [8]. The distinction between “New World” and “Old World” not only refers to the different geographic distributions but also taxonomic differences [8]. Despite these two groups being phenotypically very similar and both obligate scavengers, they are not taxonomically related to each other, and the various characteristics they share are the result of convergent evolutionary processes [8]. Their scavenging habits evolved independently, leading to adaptations such as robust bodies, broad wings, powerful beaks, and featherless heads [5,8].

### 2.2. Morpho-Functional Adaptations of Vultures to Dietary Specialization

Among existing vertebrates, vultures are the only obligate scavengers, meaning they rely strictly on carrion for their survival and reproduction [9]. To this purpose, they developed highly specialized anatomy and physiology with a series of unique adaptations. This generally includes a very sharp beak, a relatively bare head and neck, and the capability to produce strong gastric acids for digesting decomposing and contaminated meat; besides having legs adapted for ground locomotion [9]. Despite these commonalities, the 23 different species of vultures exhibit diverse phenotypes, thought to be linked to ecological differences [9]. Competition among sympatric vultures has led to differences in their feeding strategy, such as preferences for specific components of a carcass over others. Gulpers primarily feed on viscera, i.e., the softer portions of the carcass, and include most griffon species (*Gyps* spp.) and some other species like the American black vulture (*Coragyps atratus* (Bechstein, 1793)) [9]. In contrast, rippers feed mainly on the tough tissues and skin of carcasses: they include large vultures like the cinereous vulture (*Aegypius monachus* (Linnaeus, 1766)), the red-headed vulture (*Sarcogyps calvus* (Scopoli, 1786)), and the king vulture (*Sarcoramphus papa* (Linnaeus, 1758)) [9]. Scrappers primarily feed on small scraps located on the surface and around the carcass; this group includes the Egyptian vulture (*Neophron percnopterus* (Linnaeus, 1758)) and the hooded vulture (*Necrosyrtes monachus* (Temminck, 1823)) [9]. It is necessary to mention the huge lammergeier (*Gypaetus barbatus* (Linnaeus, 1758)), an osteophage vertebrate (bone-eating specialist), which ingests large pieces of the skeleton or carries and drops bones into an ossuary to break them into smaller pieces [5].

The neck characteristics include a large number of bones and joints in the cervical tract of the spine, making of vulture’s neck a structurally complex anatomical structure [9]. From a muscular point of view, this structure presents obvious complexities [9]. Gulpers are characterized by slender skulls, long necks, and bald heads, consistent with their feeding mode [1]. A slender skull and long neck may facilitate access to soft tissues deep within the various compartments of the carcass, while a bald head or neck may decrease the likelihood of infections [1]. Rippers show wider skulls and stronger beaks compared to gulpers, and feathers are generally present on the neck [1]. Hertel et al. [10] suggested that a wider skull could provide greater strength to the neck muscles involved in feeding, facilitating the removal of tough skin portions and hard fibrous tissue from the carcass; a strong beak would also be advantageous for this purpose [1]. Finally, scrappers have a slender skull and a weaker beak, unsuitable for tearing and swallowing large meat portions but suitable for pecking small portions of tissue located above and around the carcass [1]. A recent study conducted by Böhmer and colleagues [9] on a group of vultures, each representing one feeding group, demonstrated that vultures’ feeding preferences are reflected not only in skull morphology but also in the neck, limbs, and other anatomical regions [1]. Rippers use their claws to grasp and hold the carcass while feeding, while gulpers generally do not. It has been reported that rippers are capable, to some extent, of predation [1]. The ability to capture live prey varies depending on the species: white-headed vultures (*Trigonoceps occipitalis* (Burchell, 1824)) and lappet-faced vultures (*Torgos tracheliotos* (Forster, 1796)) are the species with the highest predatory capacity. This predatory ability is reflected in their gripping claws, which are used for hunting, and in the peculiar eyes of Trigonoceps. This species has a wider binocular field of view than other species, a peculiarity that can help position the claws during hunting. This predatory capacity may also be reflected in the wings, as chasing evasive prey may require a fluttering type of flight, which is atypical for vultures; they soar when searching for carcasses but generally use ascending thermal currents to do so without the need to actively flap their wings [1]. Gulpers also have some ability to capture live prey, although this occurs exceptionally [1]. Scrappers present the highest ability to capture small prey because they adopt a feeding behavior very similar to that of chickens, pecking various prey such as insects, small terrestrial vertebrates, and occasionally live fish; like rippers, they can also use their claws to anchor themselves to carcasses, although to a lesser extent [1].

The social behavior of vultures varies considerably from species to species, with some being more solitary than others [11]. New World vultures are extremely social and can gather in small groups during food searches to efficiently locate carrion, exchanging signals among themselves. From their high vantage point in the sky, most vulture species rely on sight to find food [5]. When an individual locates a carcass, it alerts its companions to a potential meal by circling it, thereby also informing other scavenging mammals like jackals and hyenas, making them a key species in the scavenging community [5]. Additionally, New World vultures of the genus Cathartes have a keen sense of smell: turkey vultures can locate carrion under dense rainforest canopies or buried under leaf litter and can lead their group mates to hidden meals they otherwise wouldn’t find [5]. Altogether, this demonstrates the ecological role of edificatory species in nature, creating favorable conditions for the habitation of other animal species [12,13]. Some species, like turkey vultures, have larger nostrils and nasal passages, a greater surface area for olfactory receptors, and relatively larger olfactory bulbs compared to black vultures [14].

### 2.3. Adaptation to the “Extreme”: From Scavenging to Drastic Temperatures

Carrion is a resource that is temporally and spatially unpredictable, which vultures are uniquely adapted to exploit. They soar in flight to search for food over vast areas, expending minimal energy, considering that migrating across areas exposes them to extreme environmental conditions that would challenge any other terrestrial animal [5]. Among the various mechanisms vultures have developed for scavenging, one of the most important for the survival of these raptors involves their ability to regulate their body temperature [15]. A rather unusual adaptation for heat management, shared by this species with the American stork, is represented by urohidrosis, where New World vultures habitually assume specific positions to defecate and urinate on their legs [5,15]. This provides instant evaporative cooling of the blood vessels in the unfeathered tarsi and legs; thanks to this technique, these animals can tolerate temperatures above 40 °C [5,15]. Following this process, their legs are stained white from uric acid. Also, urine helps to kill any bacteria or parasites that the birds may encounter while walking among carcasses or landing on them [16].

Vultures have specific regions of their bodies, including the head and neck, devoid of feathers. The reason has, traditionally, been attributed to preventing soiling when feeding on viscera. Nevertheless, recent observation reveals that these bald and feathered areas are primarily involved in thermoregulation [17]. Griffon vultures, for instance, endure greater daily temperature fluctuations than other scavenger birds [17]. Their range includes deserts with summer temperatures exceeding 40–50 °C and ground temperatures of 60–70 °C; however, these birds can take flight within minutes and reach altitudes where actual temperatures are below freezing, with some individuals even reaching altitudes of 2000 m during long-distance migrations [17]. In these conditions, there is an obvious need to conserve body heat [17] and many species of Old- and New-World vultures have adapted by employing postural changes in response to environmental temperature conditions, resulting in their large areas of bare skin being exposed in warmer conditions but covered by dense feathered areas when it is cold [17]. Feather thickness also varies significantly depending on the anatomical region considered [17,18]. The head and neck region are virtually devoid of coverage, as are the lower portions of the legs, certain areas under the wings, and two zones in the chest region. When temperatures are high, vultures adopt an expanded posture, standing more erect with their neck fully extended and wings open to dissipate heat; even the chest region is exposed, and the legs are fully extended to reveal their underside [17,19]. When temperatures are low, they tend to huddle, tucking their head and neck, devoid of feathers, under the dense plumage surrounding their neck, with the lower parts of their wings held tightly against the body and the long chest feathers covering the bare skin in this region [17]. The bird also sits on its legs so that the belly feathers cover the lower parts of the legs [17,20]. Altogether, the optimal regulation of the temperatures across the diverse environmental conditions enables the control of one of the most influencing variables on the microbial community harbored in the animal body surface (i.e., the cutaneous microbiota), with impactful effects on the overall animal health and performance [21,22].

## 3. Cutaneous Microbiota in Necrophagous Animals

The term microbiota refers to the collection of microorganisms that co-exist in the same ecological niche. Acknowledged by their ubiquitous nature, these microorganisms can colonize all anatomical districts of the organism [23]. Generally, the most represented microorganisms are bacteria, followed by fungi, protozoa, archaea, and viruses [23]. They are an integral part of the animal microbiota, forming a complex ecosystem that includes trillions of commensals, symbiotics, and even potentially pathogenic microorganisms [23]. Globally, these microorganisms establish relationships among themselves as well as with species belonging to different kingdoms, including the host, resulting in the coordinated and dynamic functionality of the superorganism [24]. Literature surveys portray the intestinal microbiota as the most investigated microbial community in all animal species; although other anatomical sections host microbial communities involved in processes of equal physiological importance [4]. In the case of vultures, for instance, the cutaneous microbiota play a fundamental role in various physiological aspects [4]. Among these, protection from the infectious agents encountered during each meal is crucial [4].

From this perspective, each vulture species features peculiar skin microbiota compositions, likely involved in singular protective mechanisms of which characterization is yet highly desirable, acknowledging the potential implications under diverse applicative fields. Altogether, state-of-the-art knowledge pinpoints that the cutaneous microbiota constitutes a barrier of microbial, physical, chemical, and immunological nature for the skin [25]. Skin microorganisms represent the first physical barrier against exogenous microorganisms of environmental origin [25]. Here, pioneering microbial specimens promote physical hindrance against the secondary colonization of pathogenic specimens, aside from establishing direct connections with the host, aimed at regulating the differentiation process of keratinocytes [26]. By employing various mechanisms of resistance to colonization, such as competition for nutritional resources, direct inhibition, and/or interference with environmental survival [25], cutaneous microbiota enable, de facto, a microbial barrier that protects from the myriads of pathogenic microorganisms encountered in necrophagic activities [4]. Intriguingly, health-promoting strains of the Staphylococcus genera adopt fine mechanisms to antagonize pathogenic strains of the same genera (e.g., quorum sensing, antibiotic production, etc.) [27,28,29]. In addition, cutaneous microorganisms degrade the triglycerides present in the sebum, releasing fatty acids [25]. This process increases the acidity of the skin, thereby hindering colonization by transient and pathogenic specimens (i.e., a chemical barrier warranted by the skin microbiota) [25]. Also, the cutaneous microbiota trigger the immune barrier by stimulating both innate and adaptive immune defenses. In this light, changes in the composition of the *Staphylococcus epidermidis* membrane are linked with the activation of T-cell-mediated immunity [30]. In a similar fashion, microbial-derived metabolites promote the release of antimicrobial peptides such as cathelicidines and beta-defensins, which have proven to be active pathogenic *Staphylococcus aureus* and *Escherichia coli* strains [31], Furthermore, metabolites of microbial origin, such as group-B vitamins, are involved in immune cell proliferation and maturation, contributing to the development of protective immunity [25,32] (Figure 1).

The skin microbiota play a crucial role in the differentiation of keratinocytes and in the formation of the skin epithelial barrier. Furthermore, the skin microbiota stimulate both innate and adaptive immune defenses, promoting the release of antimicrobial peptides, favoring the induction of neonatal tolerance, and contributing to the development of protective immunity [25].

### Composition of Cutaneous Microbiota and Main Interfering Variables

The cutaneous microbiota can be influenced by various endogenous variables (e.g., sex, species, age, genetics) and environmental factors (e.g., habitat, dietary style).

Among the endogenous variables, gender differences are mirrored as diverse hormonal patterns influencing the physical–chemical milieu where the microbial community is harbored [33]. Analogously, species and genetic differences occurring among animals, including vultures, are reflected in changing host immune systems and metabolic pathways [8]. These changing ecological niches, in turn, modulate the composition and functions of the hosted microbiota. With regards to the environmental variables, a wide variety of factors can be mentioned with dynamic impact depending on the host model and its specific stage of life. In scavenging birds, the facial microbiota is constantly exposed to a wide variety of microorganisms of both environmental and prey origin. In addition, the physical–chemical settings are dramatically changed by the dietary lifestyle, resulting in the selection of an optimized microbial community, involved in strong host–microbial interrelation, enabling the birds to thrive across the extreme conditions encountered within a narrow time span [4].

Therefore, the dietary habits of necrophagous birds can significantly alter the composition of their skin-associated microbiota [21]. As of today, a complex and diverse array of microbial specimens is commonly identified in vulture skin, whose exemplificative yet unexhaustive list is provided in Figure 2. Interestingly, the sum of microbiota-influencing variables, including the predominant role of the necrophagic dietary style, enable a biodiversity level of the skin microbiota above that observed in the gut (Figure 3).

The above composition reflects the extremely diversified cutaneous microbiota of necrophagous subjects, thus supporting their potential role in various physiological processes [4].

Vulture skin microbiota have peculiar features, making them unique compared to other non-necrophagic animals [4]. The cutaneous microbiota of two New World vulture species (*Coragyps atratus* and *Cathartes aura*) was studied by Roggenbuck and colleagues [3], who compared the facial microbiota against the intestinal counterpart. Interestingly, the study indicates that the facial microbiota of both investigated species featured a higher diversity when compared with intestinal microbiota [3]. Here, the frequent contact with diverse carcasses may explain the higher microbial diversity on the skin of the vultures, with bacteria belonging to the genera Clostridium and Fusobacterium being responsible for the greater abundance of identified Operational Taxonomic Units (OTUs) on vulture skin. Moreover, as tearing the tissues of large mammals is particularly exhausting [21], vultures often resort to natural orifices to grant access to carcass viscera, including the anal opening. This increases the likelihood of ingesting anaerobic fecal bacteria such as Clostridia and Fusobacteria [3]. Similarly, Marie Lisandra Zepeda Mendoza and colleagues [4] assessed the facial microbiota of these individuals, examining the same vulture species. Accordingly, they found a higher diversity of facial microbiota compared to intestinal, which overlapped the findings of Roggenbuck et al. [3].

A recent study conducted by Gary R. Graves and colleagues [22] explores the impact of solar irradiation, to which vultures are exposed, on the composition of the microbial community associated with the feathers of these raptors [22]. In this view, physical causes of the deterioration process affecting feathers, such as abrasion and irradiation by ultraviolet rays, have been the subject of several observational studies [34,35]. However, microorganisms residing within feathers have received little attention, although they could represent an important biological cause of such deterioration. Indeed, keratinolytic fungi and bacilli are commonly identified in the plumage of avian species [35,36]. Microbial keratinases target cross-linked structural peptides that render feather keratin water-insoluble and are thus responsible for the degradation of avian species’ plumage [37]. According to data reported in Edward H. Burtt’s study [35], the incidence of birds with keratinolytic bacilli is susceptible to changes over the seasons, with a higher load in late autumn and winter and a lower one in early spring and late summer [35,38]. Birds have developed a series of behaviors, including sunning, to safeguard the condition of their plumage [22]. Avian sunning is phylogenetically and geographically widespread and serves a variety of purposes: heat absorption and thermoregulation are the most frequently hypothesized purposes, but sun exposure often occurs at thermoneutral temperatures in tropical and temperate latitudes [22]. Birds may voluntarily sun themselves at the edge of hyperthermia, indicating a purpose other than regulating body temperature [39]. Other traditional explanations for sun exposure include drying the plumage, inhibiting feather lice, vitamin D production, and stimulating uropygial glands [39,40]. Hundreds of species of diurnal birds live in habitats devoid of vegetation that protect from solar exposure, but few species experience as much solar irradiation as New World vultures [22]. All modern birds undergo periodic molting to replace worn, degraded, and damaged feathers, with the vast majority of species undergoing at least one complete annual molt; molting temporarily reduces the load of bacteria and fungi that degrade plumage [22]. Bacterial degradation of feather keratin reduces the physical integrity of feathers, feather functionality, and the overall physical condition of individuals [35]. The replacement of primary and secondary feathers in New World vultures takes up to 2 years to complete [41,42]. The prolonged molting times in vultures impose strict selection to preserve the aerodynamic integrity of feathers, necessary for efficient and long-distance flight [22]. Frequent exposure to UV rays and associated drying and heat likely kill, inhibit growth, or induce sporulation in mesophilic bacteria and fungi sensitive to UV rays, including many that degrade keratin [43].

Altogether, only a few studies have been reported in the literature to date, leaving the role of cutaneous microbiota in necrophagous animals largely unexplored, and further research activity is desirable.

## 4. Role of Cutaneous Microbiota in Relation to Infectious Agents

One of the most intriguing aspects of vulture biology is how they protect themselves from threats posed by their food source. The literature sources analyzing the causes of this extraordinary adaptation method possessed by vultures are few and it is essential to deepen the knowledge of various biological aspects of these animals, including their vulnerability to pathogens from their diet and their role in the transmission of infectious diseases [44,45]. Vertebrate carcasses are rich in nutrient resources. It has been hypothesized that the release of toxins and pathogenicity genes in the carcass microbiome is part of a microbial strategy to outcompete other microorganisms and/or pathogens naturally present in this substrate [46]. The main colonizers of a carcass are microorganisms from the microbiota of the alive animal, some of which could become pathogens in the carcass environment [4]. Other components of the post-mortem microbiota include bacteria living in the soil, nematodes, fungi, and insects [47]. However, the complex taxonomic diversity of the microbiota and the relative gene repertoire have not yet been examined regarding their protective role against microorganisms that represent serious health risks to other non-necrophagous vertebrate species. Table 1 provides a list of the major health-promoting microorganisms of the vulture microbiota along with the potential mechanisms encompassed in their role.

To assess the protective role of facial microbiota and the intestinal microbiome in vultures, Zepeda Mendoza and colleagues [3,4], generated a set of metagenomic data using swabs from the facial and intestinal regions of two New World vulture species, the black vulture (*Coragyps atratus*) and the turkey vulture (*Cathartes aura*), performing taxonomic and functional metagenomic analyses [4]. As expected, the predominant bacterial strains of the dataset include potential human pathogens, specimens associated with bioremediation, and potentially beneficial microorganisms (e.g., producers of antibiotics, insecticides, and antifungals), typically belonging to intestinal bacteria or those related to water, plants, or soil [4], underlining the One Health relevance of studying the skin microbiota of such birds. Regarding the strict prevention from pathogen colonization, facial vulture microbiota identified *Hylemonella gracilis*, responsible for the long-term prevention of colonization by *Yersinia pestis* [48]. Also, *Lactobacillus sakei*, a common anti-listerial bacterium, is effectively identified as a skin-associated bacterium [72]. Further, the common presence of genes involved in antimicrobial compound biosynthesis suggests the effort of this microbial community in chemical-based competition, as already observed in microbial specimens inhabiting other ecological niches (e.g., soil) [4]. Nevertheless, several antimicrobial resistance traits are identified as well, suggesting a contribution of these raptors to such an alarming phenomenon [5,73,74,75,76,77,78,79,80,81,82,83,84,85,86].

## 5. The Role of Vultures in Preventing the Circulation of Infectious Diseases

Nowadays, the extremely high density of livestock, the presence of various wild animals, and inadequately organized carcass disposal sites contribute to the rising concentration of carcasses in the environment; thus, the load of pathogenic microorganisms is shed into the environment [77]. Farmers and individuals involved in livestock farming who live in close contact with animals are the most susceptible to a wide range of infectious diseases [77,82]. These spread through pathogens transferring from carcasses into soil or water and infect humans through various means, such as the consumption of contaminated water or food or direct contact [83,84]. In this scenario, vulture populations can mitigate the spread of infectious diseases in a variety of ways. Anatomically, necrophagous animals have shorter intestinal tracts than their omnivorous and herbivorous counterparts, reducing the time for pathogen multiplication introduced through ingestion and thus minimizing their presence [76,79]. Also, they feature higher immune system tolerance as compared to other animals [78]. Furthermore, serving as an environmental “clean-up” species, vultures play an active role in reducing the circulation and/or diffusion of pathogenic specimens, of importance for both the human and third animals’ health [73,74]. Here, vultures dispose of over 22% of organic waste in urban areas, contributing to an overall decrease in the incidence of infectious diseases in a One Health scenario [75]. A recent study [40] highlights the importance of scavenger species in safeguarding public health in the Indian population, focusing on zoonotic diseases such as rabies, brucellosis, and tuberculosis [40]. The study underlines that vultures can effectively regulate the spread of these diseases and reduce the sanitary costs required for treatment and prevention including the practices linked to the bonification of the environment, i.e., the most important source for the dissemination of these zoonotic pathogens [47,48].

## 6. The Role of Vultures in Antimicrobial Resistance Spread

Being migratory birds, vultures are likely to play a role in the vehiculation and spread of antibiotic-resistant bacteria, both of human and animal origin [87,88]. Indeed, feeding on livestock carcasses across large geographical areas and various continents, vultures acquire and select bacteria resistant to antimicrobial agents and transfer them across extremely distant areas, contributing to the widespread diffusion of antimicrobial resistance (AMR). This alarming capability has recently been documented by Guillermo Blanco and colleagues [89] by comparing the fecal microbiota of two different vulture species: the sedentary griffon vulture (*Gyps fulvus*) and the trans-Saharan migratory Egyptian vulture (*Neophron percnopterus*). The survey underlines high rates of AMR traits in the feces of both vulture species, suggesting these traits as possibly acquired by both the direct ingestion of resistant specimens and/or via the ingestion of antimicrobial residues from medicated livestock carcasses [89]. This evidence makes vultures a model of necrophagous animals worth keeping in mind while considering the contributors to the emergence of novel resistance clones and the prediction of AMR fluxes over long distances.

## 7. Discussion and Conclusions

Vultures contribute significantly to maintaining ecosystem homeostasis. Part of their food intake is released as nutrients through their feces, promoting a cycle of nutrients that supports a healthy ecosystem [90]. From this perspective, efforts in the characterization of the fecal microbiota would elucidate the ecological service necrophagic birds provide to other animal species. Furthermore, vultures have been recognized for their public health benefits, primarily because they can efficiently dispose of vast quantities of inadequately managed carcasses, thereby maintaining adequate levels of hygiene in the ecosystem [5]. The pivotal role of the vultures’ unique ability to contain pathogens is attributed to the microbiota associated with the different anatomical districts of their bodies, with the cutaneous microbiota as the main actor [4]. Unlike other animals, vulture skin microbiota is the most populated in terms of microbial abundance and biodiversity. This is likely due to dietary style, since carcasses carry diverse and more specialized microbial communities, interfering with skin microbial development. It can be hypothesized that the taxa identified derive from the dead animal body, such as *Methanobrevibacter smithii* present in the intestinal district of the dead animals, or from environmental sources like, for example, *Xanthomonas* spp. and *Actinobacillus pleuropneumoniae* [4]. Genes identified in the facial region include elements correlated with putrescine [47], one of the main molecules produced in a decomposing carcass, confirming the influence that the carcass microbiota has on the assembly of the host’s microbiota. Further support for this hypothesis comes from the extensive functional diversity observed in microorganisms residing in the facial region. It has been observed that these range from producers of antimicrobials to normal intestinal bacteria and bacteria linked to plants and soil [4], agreeing that there is a significant influence from the environment and carcass on the composition of the facial microbiota. In this regard, determining the carcass microbiota would be necessary for further evaluation of this hypothesis.

Regardless of the origin, the vulture’s microbiota play a protective role by directly competing with pathogenic microorganisms and contributing to the biological homeostasis of the host individual [4]. Supporting this view, several beneficial microorganisms have been identified. The identification of taxa and genes related to insecticides, fungicides, and antiparasitics in the facial microbiota suggests protective mechanisms against potential eukaryotic pathogens present in the carcass; for example, *Pseudomonas entomophila*, which causes lethality in fruit flies, might fight against eukaryotic parasites [54]. In addition to containing potentially pathogenic microbiota, carcasses also contain toxic and carcinogenic compounds, posing risks to vulture health, particularly to the skin of the facial district, which is directly exposed to such compounds [91]. PAHs are xenobiotic pollutants with negative health effects emitted by animal carcasses, previously found in high concentrations in other vulture species [92,93]. Based on the literature data, it is important to note that microbiota from the facial region are the main agents responsible for the metabolism and biodegradation of xenobiotics [4]. Additionally, bacteria that degrade phenol (from the vulture’s facial region), such as *Acinetobacter calcoaceticus*, have been found in this region [94]. Phenolic compounds can act against foodborne pathogens and bacteria responsible for the carcass decomposition process, again suggesting that these derive from carcasses, developed according to their competitive environment, rather than from the host [95]. This could suggest an opportunity to introduce better alternatives to chemical treatments used in agriculture, allowing farmers to adopt a biological approach by implementing these individuals to eliminate other harmful species to the harvest and thus reduce economic losses, producing a positive impact on the economic front and promoting a more sustainable lifestyle [75]. Scavenging animals, such as vultures, play a role in regulating the spread of various diseases. They act by reducing the density of vectors and hosts, directly feeding them, thereby reducing the concentration of pathogens in the environment [78]. Some disease agents are a huge public health problem, mainly due to the lack of a cohesive plan to manage such cases by hospitals and governments, especially in underdeveloped countries [96,97]. The presence of infected carcasses near waterways significantly increases the incidence of foodborne diseases transmitted through sources [98]. It has been found that in the absence of vultures, the number of carcasses accumulating on the ground can only increase, feeding the transmission cycle [78].

Anaerobic pathogens are the main threat to human health, as they spread from carcasses through the soil. *Clostridium perfringens*, a human anaerobic pathogen, has been isolated from water, soil, air, dust, fresh meat, milk, and vegetables [78]. It has been found that humans are more susceptible to infection from this bacterium if they are in poor hygienic conditions, similar to those promoted by abandoned carcasses in situ and inadequately managed [99]. In addition, carcasses are an easy and advantageous food source for rodents, leading to an increase in the vector population and intensifying the diseases they spread, both among wildlife and domestic and human populations [100].

Finally, in this review, we aimed to highlight the role of migratory wildlife in transferring microbial strains resistant to various classes of antimicrobial agents, providing a starting point for future monitoring of the alarming phenomenon of AMR spread. Cross-resistance associated with livestock animals is often amplified and spread by vultures, significantly impacting both public health and animal welfare [89]. New resistance determinants can easily be generated in the vulture’s microbiota, triggered by chronic exposure to varying amounts of different antimicrobials and the increasing environmental stress these animals face [101]. Here, the antimicrobial resistance reported concerns the most frequently used antimicrobials in livestock farming, particularly in intensively farmed pigs, poultry, and cattle [89,102]; these data suggest that the phenomenon of antimicrobial resistance in wildlife is always associated with the presence of anthropogenic contamination inputs in the natural habitat of these individuals [103,104].

Altogether, the skin-associated microbial community of necrophagous animals, such as the vulture model, is linked with a great variety of benefits, ranging from host health to public health and environmental benefits. Nevertheless, only a few investigations have so far been performed in this field, leaving several open questions and knowledge gaps. Filling these knowledge gaps could be of great importance in “knowledgebase discovery”, i.e., the production of fundamental knowledge with translational relevance in various fields of infectious disease management and monitoring. It could also open new avenues for research aimed at preventing the onset and spread of significant infectious diseases as well as providing suggestions for improving livestock production strategies and enhancing the health status of both the environment and wildlife.

## Figures and Tables

**Figure 1 microorganisms-13-00898-f001:**
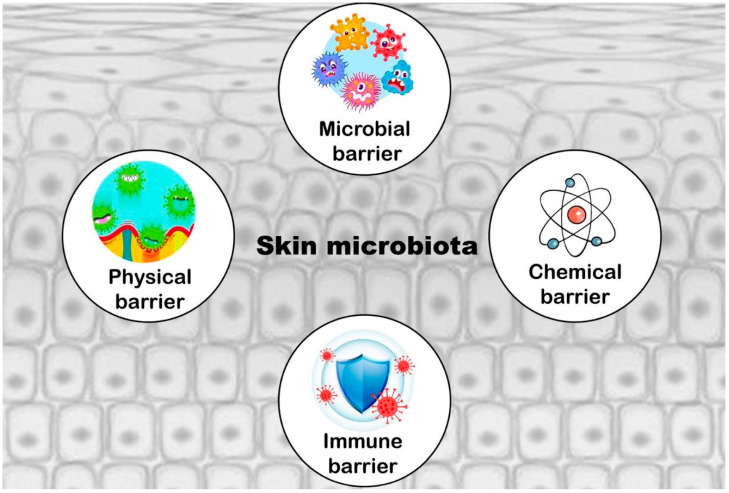
Types of barriers that the skin microbiota mediate through their functions.

**Figure 2 microorganisms-13-00898-f002:**
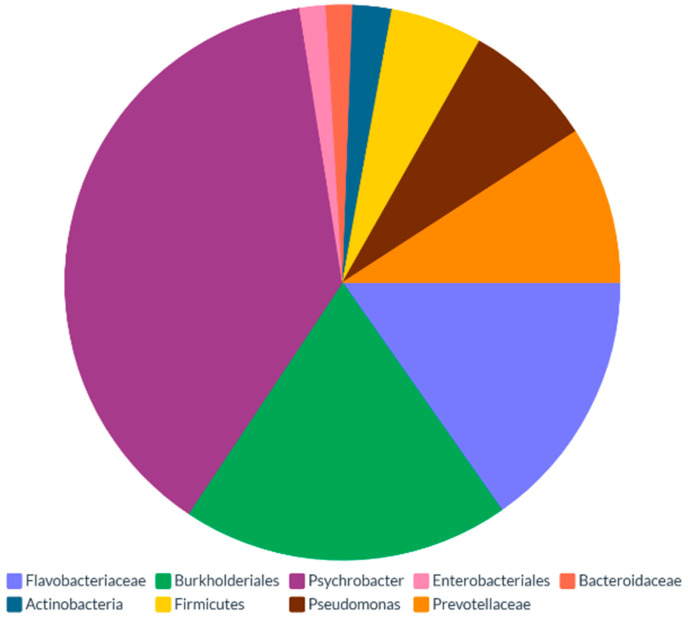
Taxonomic profile of vulture skin microbiota, as per [4].

**Figure 3 microorganisms-13-00898-f003:**
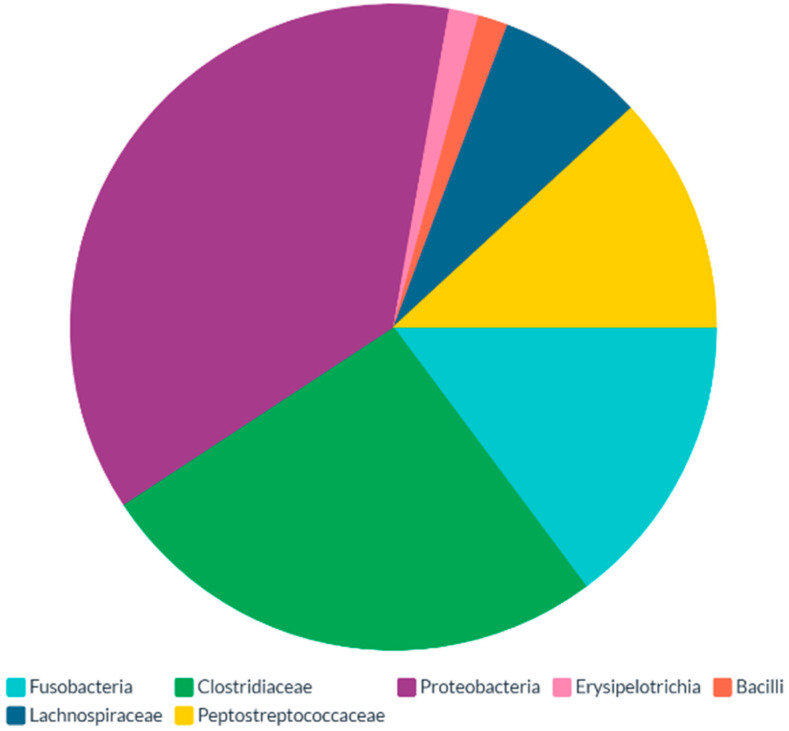
Taxonomic profile of the vulture gut microbiota, as per [4].

**Table 1 microorganisms-13-00898-t001:** Major microbial specimens of the vulture microbiota and their potential involvement in the animal’s health promotion.

Microorganisms	Health-Promoting Activity	Protective Mechanism (s)	Reference
*Hylemonella gracilis*	*Yersinia pestis* control	Prevent colonization	[48]
*Pseudomonas fluorescens*	Control of multiple pathogenic bacteria	Production of antibiotic (mupirocin), formation of protective biofilm	[49,50]
*Arthrobacter phenanthrenivorans*	Irritating substances (phenanthrene)	Degradation of phenanthrene (skin-irritating polycyclic aromatic hydrocarbon)	[51]
*Acinetobacter* sp. NIPH 899	Folate biosynthesis (potential role in skin cancer prevention)	-	[52]
*Lysinibacillus sphaericus*	Mosquito larvae	Production of insecticidal toxins (sphaericolysin)	[53]
*Pseudomonas entomophila*	Insects (fly larvae and adults)	Infection and lethality in insects, production of insecticidal toxin (SepC/Tcc class)	[54,55]
*Streptomyces violaceusniger*	Control of fungal pathogens	Antifungal activity	[56,57]
*Kitasatospora setae*	*Trichomonas* spp. control	Production of setamycin (antitrichomonal)	[58]
*Streptomyces bingchenggensis*	Control of helminths	Production of milbemycin (anthelmintic)	[59]
*Chromobacterium violaceum*	Control of multiple pathogens, tumor protection	Production of violacein (anticancer, antibacterial, antifungal, antiviral)	[60]
*Janthinobacterium* sp. HH01	Control of multiple pathogens, tumor protection	Production of violacein (anticancer, antibacterial, antifungal, antiviral)	[61]
*Polaromonas naphthalenivorans*	Naphthalene	Degradation of naphthalene (potential carcinogen)	[62]
*Yarrowia lipolytica*	Control of multiple pathogens	Production of biosurfactants (broad-spectrum antimicrobial activity)	[63]
*Rhodococcus erythropolis*	Control of multiple pathogens	Production of biosurfactants (broad-spectrum antimicrobial activity)	[64]
*Phage phi* MR11	Multidrug resistant *Staphylococcus aureus*	Lysis and clearance	[65]
*Acinetobacter* phage Petty	*Acinetobacter baumanii* control	Infection and lysis	[66]
Acibel004	*Acinetobacter baumanii* control	Infection and lysis	[67]
Phage BPP-1	*Bordetella* spp.	Infection and lysis	[68]
*Herbaspirillum frisingense*	-	Production of naphthocyclinones antibiotics	[69]
*Heterorhabditis bacteriophora*	Fleas, ants, and flies	Releasing *Photorhabdus luminescens* bacteria from their digestive tract	[70]
*Adineta vaga*	Scavenge dead bacteria and protozoans	Feeds on dead organic matter	[71]

## Data Availability

No new data were created or analyzed in this study.

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
