# Peer review of "The Role of Vulture (Accipitriformes) Cutaneous Microbiota in Infectious Disease Protection"

_microorganisms, 2025, doi:10.3390/microorganisms13040898_

Round 1
Reviewer 1 Report
Comments and Suggestions for Authors
Dear Authors,
The manuscript certainly touches upon an interesting topic. However, the connection between the vulture microbiota and disease prevention still remains under many questions. The connection with humans also needs to be expanded. The title of the manuscript should be supplemented with a clarification on taxonomy, as this clarifies the essence of the study. The text of the manuscript in different parts needs to be supplemented and rewritten. In many methodological aspects, the authors omit important information. It should be added. When comparing the microbiota of different vulture species, there is a lack of graphic material. The comparative part in the discussion needs to be expanded and additional literature sources on other species of birds of prey should be cited. The authors conducted a review of the topic of study, but did not touch on some aspects. In particular, the microbiota of vulture feces is of interest. After all the comments are eliminated, the manuscript can be reviewed again.

Author Response
The Authors gratefully acknowledge the Reviewer for the thorough revision of the manuscript and the constructive comments. The revised version of the manuscript provided is amended accordingly, along with the Reviewer1 revision file, which includes a point-to-point response.

Reviewer 2 Report
Comments and Suggestions for Authors
The present review aimed to examine the composition and functional importance of the vulture skin microbiota, exploring its role in pathogen exclusion, immune system modulation and environmental adaptation.
The manuscript is interesting, although the authors could perhaps make the review more attractive by including tables.
Some specific points are shown below.
L.211. Figure 1. It would be good to include examples of each type of barrier involved in the skin since the figure is somewhat simple.
L.213-215. This information is somewhat repetitive as it is mentioned in L.204-206.
L.249. Figure 3 is not mentioned in the text.
L.287. It would be good to include a table with the list of skin microorganisms and their effect against pathogenic bacteria and/or their toxins.
L.391. this thesis???
Author Response
The Authors acknowledge the Reviewer for the detailed revision of the manuscript. The revised version of the manuscript is amended according to this Reviewer's suggestion.

Round 2
Reviewer 1 Report
Comments and Suggestions for Authors
Dear Authors,
I am satisfied with the revision of the manuscript. Additional data and corrections have been added to the manuscript. The role of skin microbiota of vultures (Accipitriformes) in protection against infectious diseases is substantiated and convincingly presented in this manuscript. All possible aspects of interaction are revealed. The review of the research results was selected appropriately. The article takes into account the comments on the methodology. The analysis and conclusion for each chapter are sufficient and do not raise objections. References to sources of literature have been adjusted. The results of previous studies by other authors have been taken into account. I recommend it for the journal Microorganisms.